# Catholic Perspective on Decision-Making for Critically Ill Newborns and Infants

**DOI:** 10.3390/children9020207

**Published:** 2022-02-06

**Authors:** Annie B. Friedrich, Jason T. Eberl

**Affiliations:** 1Bioethics Research Center, Washington University School of Medicine, Saint Louis, MO 63110, USA; annie.friedrich@wustl.edu; 2Albert Gnaegi Center for Health Care Ethics, Saint Louis University, Saint Louis, MO 63104, USA

**Keywords:** decision-making, pediatric ethics, Catholicism, technology

## Abstract

In this paper, we discuss the foundational values informing the Catholic perspective on decision-making for critically ill newborns and infants, particularly focusing on the prudent use of medical technologies. Although the Church has consistently affirmed the general good of advances in scientific research and medicine, the technocratic paradigm of medicine may, particularly in cases with severely ill infants, lead to decision-making conflicts and breakdowns in communication between parents and providers. By exploring two paradigm cases, we offer specific practices in which providers can engage to connect with parents and avoid common technologically mediated decision-making conflicts. By focusing on the inherent relationality of all human persons, regardless of debility, and the Christian hope in the life to come, we can make decisions in the midst of the technocratic paradigm without succumbing to it.

## 1. Introduction

In 2017 and 2018, two cases involving critically ill infants garnered international attention, including that of the Vatican. Both infants, Charlie Gard and Alfie Evans, lived in the United Kingdom with their parents and were born with severe abnormalities. Eleven-month-old (at the time of his death) Charlie was diagnosed with an extremely rare genetic mutation that led to severe symptoms in multiple organs, while 23-month-old (at the time of his death) Alfie suffered from neurological degeneration due to an undiagnosed cause. In both cases, the health care team disagreed with the parents regarding whether life-sustaining treatment—as well as a potential experimental treatment, in Charlie’s case—should be continued. Charlie’s and Alfie’s respective healthcare teams argued that treatment should be discontinued insofar as it was “futile,” as well as “unkind and inhumane,” and that palliative care be provided until they died naturally [1,2]. When the U.K. courts sided with the healthcare professionals’ assessments, Charlie’s and Alfie’s parents’ battle to maintain treatment gained media traction. Eventually, however, treatment was discontinued and both infants died. Pope Francis publicly reacted to their plight: in Charlie’s case stating that he is “hoping that (his parents’) desire to accompany and care for their own child to the end is not ignored”; and in Alfie’s case appealing via Twitter “that the suffering of (Alfie’s) parents may be heard and that their desire to seek new forms of treatment may be granted.”

Yet, opinions were divided among Catholic ethicists, some of whom accused the physicians and courts of a “systemic ableist” bias against severely disabled persons [3], while others defended the decision to limit the use of treatment deemed “extraordinary [4,5,6]. This debate illuminates the foundational values informing the Catholic perspective on decision-making for critically ill newborns and infants [7,8]. These values include respect for the inherent dignity of human life, no matter how ill or impaired; respect, within limits, of the authority of parents as stewards of their children’s well-being; medicine’s vocation to alleviate suffering by morally licit means; and the prudential use of technology, avoiding both overtreatment and undertreatment.

Catholic healthcare institutions operationalize these values by distinguishing between proportionate and disproportionate treatments—the former comprising non-futile, non-experimental treatments that are expected to yield greater benefits to the patient than harms—employing forms of palliative care, and providing spiritual support for parents, families, and caregivers. Nevertheless, the pervasiveness of life-sustaining technology often interferes with shared decision-making regarding appropriate treatments [9,10,11,12]. Shared decision-making, broadly construed, can be understood as “an accepted standard of collaborative care processes that involves the patient (and family) and the clinician using the best evidence in consideration of the patient’s (and family’s) values, goals, and preferences related to decisions” [13] (p. S171). However, technological interference in shared decision-making, coupled with the lack of robust spiritual support, results in disproportionate treatments sometimes being elected out of either desperation—often couched in language of “hoping for a miracle” [14]—or a false belief that Catholic teaching requires that every means be utilized to sustain life. Conversely, when technological interventions can no longer be objectively “useful,” parents and providers may withhold or withdraw treatment based on uncertain quality of life judgments [15]. Such compromised decision-making contributes to the technocratic paradigm of modern medicine in which ersatz “liturgies” of technocratic care displace authentic forms of spiritual exercise that reinforce the Christian hope in the life to come.

## 2. Values Informing Care for Critically Ill Newborns and Infants

Foundational values, guiding principles, and authoritative directives that inform the Catholic perspective on healthcare for children have a twofold grounding in the revelation of the Hebrew and Christian Scriptures (the so-called “Old” and “New” Testaments of the Bible) and the rational discovery of the nature of human persons and the objective moral laws that should guide our relationships with one another, the rest of the natural world, and our Creator. Catholic magisterial authorities throughout the Church’s history have employed theological and philosophical forms of analysis, informed by relevant discoveries in the empirical sciences, to formulate a cohesive set of defined values that in turn inform general moral principles upon which are based more specific directives—e.g., the *Ethical and Religious Directives for Catholic Healthcare Services* promulgated by the U.S. Conference of Catholic Bishops [16]. In this section, we will briefly illuminate the combined theo-philosophical values that are most directly relevant to the care of critically ill newborns and infants. In the following section, we will discuss how the “technocratic paradigm” within which medicine typically obfuscates our understanding of these values as they inform shared decision-making on the part of parents and healthcare professionals.

### 2.1. Inherent Dignity of Human Life

In one of the earliest lines of scripture, human beings are described as having been made in the “image” and “likeness” of God (Genesis 1:26). While there has been tremendous historical and contemporary scholarly debate concerning precisely what having been created in the “imago Dei” means [17], the primary takeaway is that every individual human person has been directly willed into being by God and each of us possesses an inherent nature that shares some aspect(s) of the divine nature—e.g., human intelligence and capacity for rational thought, self-awareness, and the capacity for autonomous volition, among other putative candidates [18].

One does not have to believe, however, that such arguably essential qualities of human nature have a direct divine source. Philosophers—both Catholic and secular—have defended the idea that human beings—during all or most stages of our existence—possess intrinsic dignity: an unmeasurable and inviolable moral worth such that, as Immanuel Kant puts it, no human being can be used by another merely as a means to some other end [19,20]. Within the Catholic philosophical tradition, Thomas Aquinas contends that life is a fundamental good for human beings, as it is the basis for any other goods we may experience or bring about through our actions [21,22].

Aquinas also states, however, that life is not the only good worth preserving—e.g., love of God and the salvation of one’s soul are more important than biological life—and, thus, not every means needs to be employed to sustain one’s life [21,23]. This bivalent understanding of the value of human life has informed the development of Catholic thought concerning so-called “ordinary” (morally obligatory) and “extraordinary” (morally optional) means of preserving life—we will return to this distinction below.

### 2.2. Parental Stewardship

The primary duty to safeguard the dignified life of a newly created human being and to govern the upbringing of a child into mature adulthood lies with a child’s parents. This duty is explicitly affirmed by Aquinas as one of the fundamental moral obligations we share with other animals, who also exhibit a natural care and concern for the protection and development of their offspring [21]. Of course, we must nuance what is meant by a “parent” who bears such responsibility. While it is arguable that a child’s biological parents primarily have this duty [24], Catholic moral theology has also recognized both the validity and the inherent richness of the adoptive parental relationship [25].

The role of parents as stewards of their children’s lives and well-being has been codified into various civil laws that award parents tremendous freedom regarding how they choose to raise their children with respect to their own beliefs and values [26]. Yet, such laws also recognize the limits of parental decision-making in cases of evident abuse, neglect, or in which the parents’ beliefs and values directly threaten the life and well-being of their child [27]. In some cases, it is arguable that the state should supersede parental authority, not because the parents are making unacceptable decisions, but because the very decision at stake is one that parents should not be forced to make [28].

### 2.3. Vocation to Alleviate Suffering

It is seemingly incontrovertible that the essential goal (telos) of medicine is to alleviate suffering, though there is significant debate concerning what measures may be taken to alleviate a patient’s suffering, as seen in current debates regarding “medical aid-in-dying” and euthanasia. In at least one jurisdiction to date, the active euthanization of critically ill newborns and infants, with parental consent, is legally permitted [29]. From a Catholic perspective, such an act involves failure of healthcare professionals to realize medicine’s essential telos of healing and never harming, as well as failure on the part of parents to fulfill their fundamental stewardship obligation [30,31]. In the Gard and Evans cases, while active euthanasia was never contemplated, it was argued by their parents and supporters that Charlie and Alfie died as a result of passive euthanasia: the withdrawal of life-sustaining treatment and withholding (in Charlie’s case) of a potentially curative experimental treatment.

As noted above, the Catholic Church does not teach that any withholding or withdrawal of life-sustaining treatment constitutes illicit passive euthanasia. Rather, a distinction is made between treatment deemed to be “ordinary” insofar as it is (a) part of the available standard of care, (b) does not entail inordinate personal economic burden, (c) is not physiologically futile, and (d) entails burdens that are proportionate to the expected benefits to the patient, and treatment that is deemed “extraordinary” because it fails to meet one or more of these conditions [32]. While healthcare professionals and parents must ensure the provision of ordinary treatments under their stewardship obligation, extraordinary treatments are morally optional and, in some cases, may become morally illicit by causing a significant degree of suffering for the child with limited hope of benefit.

There is no definitive list of ordinary versus extraordinary treatments; any treatment may be ordinary for one patient but extraordinary for another due to their underlying health condition and other pertinent circumstances. Furthermore, this distinction may change for the same patient over time as their condition changes. Some specific interventions, however, have been recently deemed to be ordinary in typical cases, while allowing for a limited set of cases in which they might be deemed extraordinary [33,34,35]. Appropriate decision-making in this context requires the cultivation of the intellectual and moral virtue of prudence: a disposition toward reasoning well about how to apply general moral principles to concrete particular situations [36]. This virtue is all the more necessary given the exponential growth of various life-sustaining technologies and treatments.

### 2.4. Prudential Use of Technology

Despite a generally ahistorical view of the Catholic Church standing against scientific advancement, the Church has consistently affirmed the general good of advances in scientific research and medicine, with Catholic healthcare in the U.S. currently comprising a significant market-share, a legacy of the hospitals instituted in the 19th and early 20th centuries and initially staffed by Catholic religious sisters [37,38]. Yet, the Church notes that “every technical advance in healthcare calls for growth in moral discernment to avoid an unbalanced and dehumanizing use of the technologies especially in the critical or terminal states of human life” [35]. The application of divinely-bestowed human intelligence—a key feature of human beings’ having been created in the imago Dei—to scientific inquiry and technological development must always have as its telos the objective flourishing of individual human beings, balanced with the common good of humanity and the rest of creation.

A key concern, however, that has been voiced by secular and Christian philosophers alike, is that a technology has the inherent power to “enframe” [39] human thought and value systems such that we become a “technological society” [40]. This concern has entailed an anthropocentric devalorization of the rest of the natural world, as well as a reduction of human beings to a “standing reserve” of instruments “ready-to-hand” for exploitation [39]. All this has led to the emergence of a “use and throwaway culture” dominated by a “technocratic paradigm” in which all problems—social, political, biomedical, etc.—are viewed as resolvable through the mere application of more technology and technocratic/bureaucratic processes without regard for any overarching *telos* [41]. Thus, as Don Ihde aptly notes, we must recognize that we live in a “technologically textured world,” and examine critically how technology shapes our lives in morally meaningful ways [42] (p. 1). Simply because of the familiarity of technology, “we may overlook both the need for and the results to be obtained by a critical reflection upon our lives within this technologically textured ecosystem” [42] (p. 3).

As Pope Francis recognizes, overcoming the technocratic paradigm will require a radical shift in thinking:

The idea of promoting a different cultural paradigm and employing technology as a mere instrument is nowadays inconceivable. The technological paradigm has become so dominant that it would be difficult to do without its resources and even more difficult to utilize them without being dominated by their internal logic.[41] (n. 108)

This is why the technological imperative, or the idea that what can be done must be done, is so difficult to resist. [43] Yet, there may be hope for parents and healthcare providers to develop a prudent attitude toward the use of biomedical technology. First, though, we need to examine predominant imprudent attitudes toward biomedical technology.

## 3. Imprudent Attitudes toward Biomedical Technology

Hospitals are technologically and technocratically dominated environments, and this is no more evident than in intensive care settings, such as those in which Charlie and Alfie received care. The highly technologized character of intensive care units (ICUs) can be seen as part of what Albert Borgmann calls the “device paradigm,” the systematic and patterned character of technology that shapes our lives in damaging—and often unexamined—ways [44]. In intensive care settings, especially pediatric or neonatal ICUs, the routine and often morally licit means of preserving life require reliance on devices such as ventilators and feeding tubes; but, what may be seen as morally required and ordinary technological care may also lead to eventual disengagement from one’s child. The device paradigm of pediatric ICUs also disrupts shared decision-making between parents and providers, although this disruption is often insidious and easily overlooked. Specific technological interventions not only enter into the decision-making process and circumscribe what decisions can and cannot be made, but the technological milieu itself shapes the decision-making process, and parents and providers are not sufficiently attuned to the ways in which technologies enter into the decision-making calculus.

Let us consider the more obvious ways in which technology impacts decision-making. First, certain decision points are created by the technologies themselves, such as when parents and providers must decide when to remove ventilator support or whether to provide a gastric tube for feeding. In Gard’s case, the technological possibility of experimental treatment led to the conflict between his parents and providers: should clinicians offer the experimental treatment given the possibility that Charlie will experience significant physical and cognitive limitations in the future, or might further treatment mitigate these harmful effects? Second, parents and providers may rely on technologies to make decisions or provide answers to their questions, such as when certain MRI results lead to the conclusion—or at least suggestion—that a child’s deficits will be severe, her life will “not be worth living,” and thus life-support should be withdrawn.

It is important to emphasize that general “quality of life” assessments are excluded from a Catholic moral analysis given the Church’s affirmation of the inherent dignity of all human life regardless of disease or disability; nevertheless, understanding a patient’s prognosis in light of available treatment options, each of which may present their own burdens or risk of harm, may legitimately inform the assessment of whether a particular treatment may be construed as ordinary/proportionate or extraordinary/disproportionate. This assessment may be particularly challenging in pediatric contexts, in which children cannot voice their own values or make claims as to what counts as sufficiently “burdensome.” Without the ability to draw on explicit patient preferences, parents and providers increasingly rely on technologies (either explicitly or implicitly) to help them make these types of value-laden judgments.

Perhaps most notably, the device paradigm often robs parents of decision-making power altogether. In a qualitative study of parental decision-making experiences in the NICU, one mother said:

I mean, it’s like, “Oh, well, we had to do this because this is what was going on, and if we didn’t, this is what could happen.” You know? So I mean, there really almost wasn’t a decision to be made because it’s like, well, it had to be done.

Another mother said:

So many decisions are just made for you. In the NICU, you don’t feel like you’re really the one who gets to make those final decisions.

For many parents, this lack of decision-making power contributes to feelings of alienation from their child and from their parental role [45].

The device paradigm of the ICU also inhibits shared decision-making by masking one’s values under the guise of technological capacity, which often leads to stalemates between dissenting parties. For example, as seen in the Gard and Evans cases, parents may want to continue aggressive treatments or technological interventions to keep their infant alive—supporting their underlying value that their child’s life is worth living and that they will not give up on their child, perhaps “hoping for a miracle”—while providers may argue that this continued treatment is causing the infant unnecessary pain and suffering—supporting their underlying value that this child will have a poor quality of life and thus further interventions are inappropriate. The decision is not actually about the technology; rather, it is about the values underlying one’s claim to technologies. Yet, it is much easier to talk about specific technological interventions, so these technologies become the scapegoat for difficult discussions in pediatric intensive care settings and muddy the waters of shared decision-making.

The technological capacity to intervene shrouds the competing underlying values at stake in the Gard and Evans cases, as well. On the one hand, Charlie’s and Alfie’s parents believe that their sons are a gift from God and will love them no matter their cognitive or physical limitations—in alignment with the Catholic view of every human being’s intrinsic dignity—but they may also be suppressing a deep fear of being responsible for their son’s death if they authorize a transition to comfort care. This view of comfort care does not necessarily align with Catholic teaching so long as a transition to comfort care is a refusal of extraordinary treatment. On the other hand, providers may believe that Charlie’s or Alfie’s life is not worth living after seeing other babies with similar issues suffer—an overall quality-of-life consideration inimical to the Catholic moral perspective. When technologies can no longer be objectively useful—i.e., cannot restore the child to a specific level of function—providers may make claims about “quality of life” and “suffering” that reveal their immersion in the device paradigm, in which physical and social engagement with things—or in this case, sick infants—is burdensome rather than meaningful. Such an attitude can then lead to the determination that further treatment would be “inappropriate” or “non-beneficial” because the patient’s overall condition is not expected to improve [46]. Technological interventions thus introduce further complexity into an already-difficult decision-making process, and both parents and providers may be ill-equipped to articulate and explore the values implicitly informing their decisions, which may lead to a breakdown in the decision-making process and a stalemate between parents and providers.

## 4. Fostering Shared Decision-Making

In order to avoid a breakdown in shared decision-making in high-tech medical spaces, providers, particularly those working in Catholic healthcare, can engage in practices to connect with parents and avoid common technologically mediated decision-making conflicts. First, providers can ensure that parents have access to sufficient spiritual support throughout their child’s hospitalization, not just when they are facing a difficult decision. Having a hospitalized child is extremely trying for parents and families, and the emotional impact is not limited to the discrete period of the child’s hospitalization. Several studies have found that the neonatal ICU (NICU) experience in particular is associated with acute and post-traumatic stress disorders that continue even after the child has been discharged [47,48]. These adverse parental stress reactions can be exacerbated when their spiritual needs are not acknowledged. Parents are often drawn to spirituality to help cope with their child’s illness, and parents of seriously or terminally ill children have expressed a need for spiritual support [49,50,51]. One study found that mothers who received a direct spiritual care intervention while their child was in the NICU had less stress than mothers who did not receive the intervention [52]. By directly engaging in spiritual care throughout the child’s hospitalization, providers can better understand the family’s values and situate decisions within these values, ultimately striving to avoid what are sometimes perceived as “bad” decisions grounded in tenuous and unexplored values.

Second, providers ought to cultivate a prudential use of technology in their practice. While we suggest that parents and providers ought to be critical of technology and its debilitating and disengaging effects, critically analyzing the technocratic or device paradigm does not necessarily entail ridding oneself or one’s practice of technology completely. On the contrary, when medical staff and parents explicitly discuss the technological elements of the child’s care in relation to shared values and goals, the clinical relationship to technology can become more clarified and well-defined. In order to bring about this proper relation to technology, clinicians can intentionally discuss and explain certain technological interventions and machinery used in the child’s care—even seemingly “mundane” ones—in order to draw parents into this often-unfamiliar and overwhelming place. Clinicians can also turn off monitors or machines when possible and can encourage parents to touch, hold, and engage with their baby whenever possible.

## 5. Conclusions

Ultimately, in order to push back against the damaging effects of the technological character of modern medicine, parents and providers must specifically address technology’s role in the decision-making process and be more intentional with their use of technology. Our aim is not to disrupt or do away with the techniques and technologies we currently use to care for sick infants; rather, we argue that parents and providers can potentially avoid common decision-making conflicts (or can at least come to a better shared understanding of the situation) by explicitly situating and discussing goals and values in light of the profound yet subtle influence of the technocratic paradigm of modern medicine.

Of course, we cannot directly assess the particular factors that led to the breakdown in shared decision-making between Charlie Gard’s and Alfie Evans’s parents and healthcare providers. Nevertheless, the significantly debilitated nature of their respective conditions, which required extensive technological support, lends credence to our conclusion that at least one foundational element of their disagreement was differing views of the role of technology in sustaining these young lives—whether such technology is being viewed as disproportionately burdensome or as sustaining the life of a dignified, albeit severely debilitated, human being. While attuning to the ways in which the technocratic paradigm shapes these cases may not actually change the outcome, it allows parents and providers to work together when facing inevitable suffering instead of pitting each other as the enemy. Perhaps, by focusing on the inherent relationality of all human persons, regardless of debility, and the Christian hope in the life to come, we can make decisions in the midst of the technocratic paradigm without succumbing to it.

## Data Availability

Not applicable.

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
