# Peer review of "Catholic Perspective on Decision-Making for Critically Ill Newborns and Infants"

_children, 2022, doi:10.3390/children9020207_

Round 1
Reviewer 1 Report
This is an excellent article that builds on two case studies in order to demonstrate the complexity involved in decision-making processes when it comes to critically ill infants and newborns. The article also has implications for the larger issue of the impact of technology on decision-making processes in medical practice in general. The article is well-structured, coherent, and quite easy to follow and the authors demonstrate an awareness for the complexity of issues involved. The references are well-chosen and complement the main text meaningfully.
My only suggestion is that the article could tone-down the usage of terms like "technocratic paradigm", "priesthood of technology" and the like, especially in the later part of the article, where the argument is more nuanced and the emphasis should be on that and not on such "buzz-words". I have suggested a way in which the authors can do that (see the comments in the attached document), but it is entirely up to them to decide whether to follow this suggestion.

Reviewer 2 Report
The paper gives a good overview of Catholic teachings and positions on end-of-life decisionmaking in critically ill newborns and infants. People often have some missconceptions on Cahtolic techings regardin the end of life. This paper tries to put things in clear perspective.
Author Response
My co-author and I thank the reviewer for their positive appraisal of our paper!